# FREEMATCH: SELF-ADAPTIVE THRESHOLDING FOR SEMI-SUPERVISED LEARNING

**Yidong Wang**[1,2,][\*] **Hao Chen**[3][\*], **Qiang Heng**[4], **Wenxin Hou**[5], **Yue Fan**[6],
**Zhen Wu**[7], **Jindong Wang**[1,][†] **Marios Savvides**[3], **Takahiro Shinozaki**[2],
**Bhiksha Raj**[3,8], **Bernt Schiele**[6], **Xing Xie**[1]

[1]Microsoft Research Asia, [2]Tokyo Institute of Technology, [3]Carnegie Mellon University,
[4]North Carolina State University, [5]Microsoft STCA,
[6]Max Planck Institute for Informatics, Saarland Informatics Campus,
[7]Nanjing University, [8]Mohamed bin Zayed University of AI

## ABSTRACT

Semi-supervised Learning (SSL) has witnessed great success owing to the impressive performances brought by various methods based on pseudo labeling and consistency regularization. However, we argue that existing methods might fail to utilize the unlabeled data more effectively since they either use a pre-defined / fixed threshold or an ad-hoc threshold adjusting scheme, resulting in inferior performance and slow convergence. We first analyze a motivating example to obtain intuitions on the relationship between the desirable threshold and model's learning status. Based on the analysis, we hence propose *FreeMatch* to adjust the confidence threshold in a self-adaptive manner according to the model's learning status. We further introduce a self-adaptive class fairness regularization penalty to encourage the model for diverse predictions during the early training stage. Extensive experiments indicate the superiority of FreeMatch especially when the labeled data are extremely rare. FreeMatch achieves **5.78**%, **13.59**%, and **1.28**% error rate reduction over the latest state-of-the-art method FlexMatch on CIFAR-10 with 1 label per class, STL-10 with 4 labels per class, and ImageNet with 100 labels per class, respectively. Moreover, FreeMatch can also boost the performance of imbalanced SSL. The codes can be found at `https://github.com/microsoft/Semi-supervised-learning`.[1]

## 1 INTRODUCTION

The superior performance of deep learning heavily relies on supervised training with sufficient labeled data (He et al., 2016; Vaswani et al., 2017; Dong et al., 2018). However, it remains laborious and expensive to obtain massive labeled data. To alleviate such reliance, semi-supervised learning (SSL) (Zhu, 2005; Zhu & Goldberg, 2009; Sohn et al., 2020; Rosenberg et al., 2005; Gong et al., 2016; Kervadec et al., 2019; Dai et al., 2017) is developed to improve the model's generalization performance by exploiting a large volume of unlabeled data. Pseudo labeling (Lee et al., 2013; Xie et al., 2020b; McLachlan, 1975; Rizve et al., 2020) and consistency regularization (Bachman et al., 2014; Samuli & Timo, 2017; Sajjadi et al., 2016) are two popular paradigms designed for modern SSL. Recently, their combinations have shown promising results (Xie et al., 2020a; Sohn et al., 2020; Pham et al., 2021; Xu et al., 2021; Zhang et al., 2021). The key idea is that the model should produce similar predictions or the same pseudo labels for the same unlabeled data under different perturbations following the smoothness and low-density assumptions in SSL (Chapelle et al., 2006).

A potential limitation of these threshold-based methods is that they either need a *fixed threshold* (Xie et al., 2020a; Sohn et al., 2020; Zhang et al., 2021; Guo & Li, 2022) or an *ad-hoc threshold adjusting*

---

[\*]Equal Contribution: yidongwang37@gmail.com, haoc3@andrew.cmu.edu; work done when Yidong was a research intern at MSRA.

[†]Correspondence to: jindong.wang@microsoft.com

[1]Note the results of this paper are obtained using TorchSSL (Zhang et al., 2021). We also provide codes and logs in USB (Wang et al., 2022).

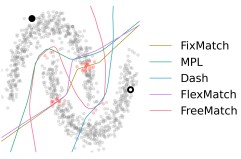 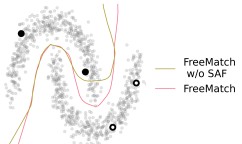 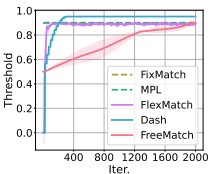 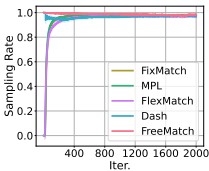

| (a) Decision boundary | (b) Self-adaptive fairness | (c) Confi. threshold | (d) Sampling rate |

Figure 1: Demonstration of how FreeMatch works on the "two-moon" dataset. (a) Decision boundary of FreeMatch and other SSL methods. (b) Decision boundary improvement of self-adaptive fairness (SAF) on two labeled samples per class. (c) Class-average confidence threshold. (d) Class-average sampling rate of FreeMatch during training. The experimental details are in Appendix A.

scheme (Xu et al., 2021) to compute the loss with only confident unlabeled samples. Specifically, UDA (Xie et al., 2020a) and FixMatch (Sohn et al., 2020) retain a fixed high threshold to ensure the quality of pseudo labels. However, a fixed high threshold ($0.95$) could lead to low data utilization in the early training stages and ignore the different learning difficulties of different classes. Dash (Xu et al., 2021) and AdaMatch (Berthelot et al., 2022) propose to gradually grow the fixed *global* (dataset-specific) threshold as the training progresses. Although the utilization of unlabeled data is improved, their ad-hoc threshold adjusting scheme is arbitrarily controlled by hyper-parameters and thus disconnected from model's learning process. FlexMatch (Zhang et al., 2021) demonstrates that different classes should have different *local* (class-specific) thresholds. While the local thresholds take into account the learning difficulties of different classes, they are still mapped from a *pre-defined fixed* global threshold. Adsh (Guo & Li, 2022) obtains adaptive thresholds from a pre-defined threshold for imbalanced Semi-supervised Learning by optimizing the the number of pseudo labels for each class. In a nutshell, these methods might be incapable or insufficient in terms of adjusting thresholds according to model's learning progress, thus impeding the training process especially when labeled data is too scarce to provide adequate supervision.

For example, as shown in Figure 1(a), on the "two-moon" dataset with only 1 labeled sample for each class, the decision boundaries obtained by previous methods fail in the low-density assumption. Then, two questions naturally arise: *1) Is it necessary to determine the threshold based on the model learning status?* and *2) How to adaptively adjust the threshold for best training efficiency?*

In this paper, we first leverage a motivating example to demonstrate that different datasets and classes should determine their global (dataset-specific) and local (class-specific) thresholds based on the model's learning status. Intuitively, we need a low global threshold to utilize more unlabeled data and speed up convergence at early training stages. As the prediction confidence increases, a higher global threshold is necessary to filter out wrong pseudo labels to alleviate the confirmation bias (Arazo et al., 2020). Besides, a local threshold should be defined on each class based on the model's confidence about its predictions. The "two-moon" example in Figure 1(a) shows that the decision boundary is more reasonable when adjusting the thresholds based on the model's learning status.

We then propose *FreeMatch* to adjust the thresholds in a *self-adaptive* manner according to learning status of each class (Guo et al., 2017). Specifically, FreeMatch uses the self-adaptive thresholding (SAT) technique to estimate both the global (dataset-specific) and local thresholds (class-specific) via the exponential moving average (EMA) of the unlabeled data confidence. To handle barely supervised settings (Sohn et al., 2020) more effectively, we further propose a class fairness objective to encourage the model to produce fair (i.e., diverse) predictions among all classes (as shown in Figure 1(b)). The overall training objective of FreeMatch maximizes the mutual information between model's input and output (John Bridle, 1991), producing confident and diverse predictions on unlabeled data. Benchmark results validate its effectiveness. To conclude, our contributions are:

- Using a motivating example, we discuss *why* thresholds should reflect the model's learning status and provide some intuitions for designing a threshold-adjusting scheme.

- We propose a novel approach, FreeMatch, which consists of Self-Adaptive Thresholding (SAT) and Self-Adaptive class Fairness regularization (SAF). SAT is a threshold-adjusting scheme that is *free* of setting thresholds manually and SAF encourages diverse predictions.

- Extensive results demonstrate the superior performance of FreeMatch on various SSL benchmarks, especially when the number of labels is very limited (e.g, an error reduction of **5.78%** on CIFAR-10 with 1 labeled sample per class).

## 2 A MOTIVATING EXAMPLE

In this section, we introduce a binary classification example to motivate our threshold-adjusting scheme. Despite the simplification of the actual model and training process, the analysis leads to some interesting implications and provides insight into how the thresholds should be set.

We aim to demonstrate the necessity of the self-adaptability and increased granularity in confidence thresholding for SSL. Inspired by (Yang & Xu, 2020), we consider a binary classification problem where the true distribution is an even mixture of two Gaussians (i.e., the label $Y$ is equally likely to be positive $(+1)$ or negative $(-1)$). The input $X$ has the following conditional distribution:

$$X \mid Y = -1 \sim \mathcal{N}(\mu_1, \sigma_1^2), X \mid Y = +1 \sim \mathcal{N}(\mu_2, \sigma_2^2). \tag{1}$$

We assume $\mu_2 > \mu_1$ without loss of generality. Suppose that our classifier outputs confidence score $s(x) = 1/[1+\exp(-\beta(x-\frac{\mu_1+\mu_2}{2}))]$, where $\beta$ is a positive parameter that reflects the model learning status and it is expected to gradually grow during training as the model becomes more confident. Note that $\frac{\mu_1+\mu_2}{2}$ is in fact the Bayes' optimal linear decision boundary. We consider the scenario where a fixed threshold $\tau \in (\frac{1}{2}, 1)$ is used to generate pseudo labels. A sample $x$ is assigned pseudo label $+1$ if $s(x) > \tau$ and $-1$ if $s(x) < 1 - \tau$. The pseudo label is 0 (masked) if $1 - \tau \leq s(x) \leq \tau$.

**We then derive the following theorem to show the necessity of self-adaptive threshold:**

**Theorem 2.1.** *For a binary classification problem as mentioned above, the pseudo label $Y_p$ has the following probability distribution:*

$$P(Y_p = 1) = \frac{1}{2}\Phi(\frac{\frac{\mu_2-\mu_1}{2} - \frac{1}{\beta}\log(\frac{\tau}{1-\tau})}{\sigma_2}) + \frac{1}{2}\Phi(\frac{\frac{\mu_1-\mu_2}{2} - \frac{1}{\beta}\log(\frac{\tau}{1-\tau})}{\sigma_1}),$$

$$P(Y_p = -1) = \frac{1}{2}\Phi(\frac{\frac{\mu_2-\mu_1}{2} - \frac{1}{\beta}\log(\frac{\tau}{1-\tau})}{\sigma_1}) + \frac{1}{2}\Phi(\frac{\frac{\mu_1-\mu_2}{2} - \frac{1}{\beta}\log(\frac{\tau}{1-\tau})}{\sigma_2}), \tag{2}$$

$$P(Y_p = 0) = 1 - P(Y_p = 1) - P(Y_p = -1),$$

*where $\Phi$ is the cumulative distribution function of a standard normal distribution. Moreover, $P(Y_p = 0)$ increases as $\mu_2 - \mu_1$ gets smaller.*

The proof is offered in Appendix B. Theorem 2.1 has the following implications or interpretations:

(i) Trivially, unlabeled data utilization (sampling rate) $1 - P(Y_p = 0)$ is directly controlled by threshold $\tau$. As the confidence threshold $\tau$ gets larger, the unlabeled data utilization gets lower. At early training stages, adopting a high threshold may lead to low sampling rate and slow convergence since $\beta$ is still small.

(ii) More interestingly, $P(Y_p = 1) \neq P(Y_p = -1)$ if $\sigma_1 \neq \sigma_2$. In fact, the larger $\tau$ is, the more imbalanced the pseudo labels are. This is potentially undesirable in the sense that we aim to tackle a balanced classification problem. Imbalanced pseudo labels may distort the decision boundary and lead to the so-called pseudo label bias. An easy remedy for this is to use class-specific thresholds $\tau_2$ and $1 - \tau_1$ to assign pseudo labels.

(iii) The sampling rate $1 - P(Y_p = 0)$ decreases as $\mu_2 - \mu_1$ gets smaller. In other words, the more similar the two classes are, the more likely an unlabeled sample will be masked. As the two classes get more similar, there would be more samples mixed in feature space where the model is less confident about its predictions, thus a moderate threshold is needed to balance the sampling rate. Otherwise we may not have enough samples to train the model to classify the already difficult-to-classify classes.

The intuitions provided by Theorem 2.1 is that at the early training stages, $\tau$ should be low to encourage diverse pseudo labels, improve unlabeled data utilization and fasten convergence. However, as training continues and $\beta$ grows larger, a consistently low threshold will lead to unacceptable confirmation bias. Ideally, the threshold $\tau$ should increase along with $\beta$ to maintain a stable sampling rate throughout. Since different classes have different levels of intra-class diversity (different $\sigma$) and some classes are harder to classify than others ($\mu_2 - \mu_1$ being small), a fine-grained *class-specific* threshold is desirable to encourage fair assignment of pseudo labels to different classes. The challenge is how to design a threshold adjusting scheme that takes all implications into account, which is

the main contribution of this paper. We demonstrate our algorithm by plotting the average threshold trend and marginal pseudo label probability (i.e. sampling rate) during training in Figure 1(c) and 1(d). To sum up, we should determine global (dataset-specific) and local (class-specific) thresholds by estimating the learning status via predictions from the model. Then, we detail FreeMatch.

## 3 PRELIMINARIES

In SSL, the training data consists of labeled and unlabeled data. Let $\mathcal{D}_L = \{(x_b, y_b) : b \in [N_L]\}$ and $\mathcal{D}_U = \{u_b : b \in [N_U]\}^2$ be the labeled and unlabeled data, where $N_L$ and $N_U$ is their number of samples, respectively. The supervised loss for labeled data is:

$$\mathcal{L}_s = \frac{1}{B} \sum_{b=1}^{B} \mathcal{H}(y_b, p_m(y|\omega(x_b))), \tag{3}$$

where $B$ is the batch size, $\mathcal{H}(\cdot, \cdot)$ refers to cross-entropy loss, $\omega(\cdot)$ means the stochastic data augmentation function, and $p_m(\cdot)$ is the output probability from the model.

For unlabeled data, we focus on pseudo labeling using cross-entropy loss with confidence threshold for entropy minimization. We also adopt the "Weak and Strong Augmentation" strategy introduced by UDA (Xie et al., 2020a). Formally, the unsupervised training objective for unlabeled data is:

$$\mathcal{L}_u = \frac{1}{\mu B} \sum_{b=1}^{\mu B} \mathbb{1}(\max(q_b) > \tau) \cdot \mathcal{H}(\hat{q}_b, Q_b). \tag{4}$$

We use $q_b$ and $Q_b$ to denote abbreviation of $p_m(y|\omega(u_b))$ and $p_m(y|\Omega(u_b))$, respectively. $\hat{q}_b$ is the hard "one-hot" label converted from $q_b$, $\mu$ is the ratio of unlabeled data batch size to labeled data batch size, and $\mathbb{1}(\cdot > \tau)$ is the indicator function for confidence-based thresholding with $\tau$ being the threshold. The weak augmentation (i.e., random crop and flip) and strong augmentation (i.e., RandAugment Cubuk et al. (2020)) is represented by $\omega(\cdot)$ and $\Omega(\cdot)$ respectively.

Besides, a fairness objective $\mathcal{L}_f$ is usually introduced to encourage the model to predict each class at the same frequency, which usually has the form of $\mathcal{L}_f = \mathbf{U} \log \mathbb{E}_{\mu B} [q_b]$ (Andreas Krause, 2010), where $\mathbf{U}$ is a uniform prior distribution. One may notice that using a uniform prior not only prevents the generalization to non-uniform data distribution but also ignores the fact that the underlying pseudo label distribution for a mini-batch may be imbalanced due to the sampling mechanism. The uniformity across a batch is essential for fair utilization of samples with per-class threshold, especially for early-training stages.

## 4 FREEMATCH

### 4.1 SELF-ADAPTIVE THRESHOLDING

We advocate that the key to determining thresholds for SSL is that thresholds should reflect the learning status. The learning effect can be estimated by the prediction confidence of a well-calibrated model (Guo et al., 2017). Hence, we propose *self-adaptive thresholding* (SAT) that automatically defines and adaptively adjusts the confidence threshold for each class by leveraging the model predictions during training. SAT first estimates a global threshold as the EMA of the confidence from the model. Then, SAT modulates the global threshold via the local class-specific thresholds estimated as the EMA of the probability for each class from the model. When training starts, the threshold is low to accept more possibly correct samples into training. As the model becomes more confident, the threshold adaptively increases to filter out possibly incorrect samples to reduce the confirmation bias. Thus, as shown in Figure 2, we define SAT as $\tau_t(c)$ indicating the threshold for class $c$ at the $t$-th iteration.

**Self-adaptive Global Threshold** We design the global threshold based on the following two principles. First, the global threshold in SAT should be related to the model's confidence on unlabeled

---

$^2[N] := \{1, 2, \ldots, N\}.$

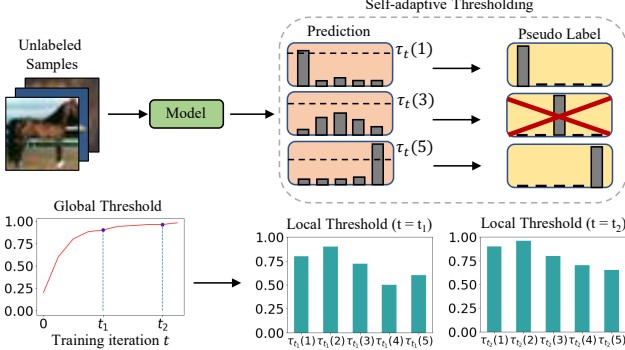

Figure 2: Illustration of Self-Adaptive Thresholding (SAT). FreeMatch adopts both global and local self-adaptive thresholds computed from the EMA of prediction statistics from unlabeled samples. Filtered (masked) samples are marked with red X.

data, reflecting the overall learning status. Moreover, the global threshold should stably increase during training to ensure incorrect pseudo labels are discarded. We set the global threshold $\tau_t$ as average confidence from the model on unlabeled data, where $t$ represents the $t$-th time step (iteration). However, it would be time-consuming to compute the confidence for all unlabeled data at every time step or even every training epoch due to its large volume. Instead, we estimate the global confidence as the exponential moving average (EMA) of the confidence at each training time step. We initialize $\tau_t$ as $\frac{1}{C}$ where $C$ indicates the number of classes. The global threshold $\tau_t$ is defined and adjusted as:

$$\tau_t = \begin{cases} \frac{1}{C}, & \text{if } t = 0, \\ \lambda\tau_{t-1} + (1-\lambda)\frac{1}{\mu B}\sum_{b=1}^{\mu B}\max(q_b), & \text{otherwise,} \end{cases} \tag{5}$$

where $\lambda \in (0, 1)$ is the momentum decay of EMA.

**Self-adaptive Local Threshold** The local threshold aims to modulate the global threshold in a class-specific fashion to account for the intra-class diversity and the possible class adjacency. We compute the expectation of the model's predictions on each class $c$ to estimate the class-specific learning status:

$$\tilde{p}_t(c) = \begin{cases} \frac{1}{C}, & \text{if } t = 0, \\ \lambda\tilde{p}_{t-1}(c) + (1-\lambda)\frac{1}{\mu B}\sum_{b=1}^{\mu B} q_b(c), & \text{otherwise,} \end{cases} \tag{6}$$

where $\tilde{p}_t = [\tilde{p}_t(1), \tilde{p}_t(2), \ldots, \tilde{p}_t(C)]$ is the list containing all $\tilde{p}_t(c)$. Integrating the global and local thresholds, we obtain the final self-adaptive threshold $\tau_t(c)$ as:

$$\tau_t(c) = \text{MaxNorm}(\tilde{p}_t(c)) \cdot \tau_t = \frac{\tilde{p}_t(c)}{\max\{\tilde{p}_t(c) : c \in [C]\}} \cdot \tau_t, \tag{7}$$

where $\text{MaxNorm}$ is the Maximum Normalization (i.e., $x' = \frac{x}{\max(x)}$). Finally, the unsupervised training objective $\mathcal{L}_u$ at the $t$-th iteration is:

$$\mathcal{L}_u = \frac{1}{\mu B}\sum_{b=1}^{\mu B}\mathbb{1}(\max(q_b) > \tau_t(\arg\max(q_b))) \cdot \mathcal{H}(\hat{q}_b, Q_b). \tag{8}$$

## 4.2 Self-Adaptive Fairness

We include the class fairness objective as mentioned in Section 3 into FreeMatch to encourage the model to make diverse predictions for each class and thus produce a meaningful self-adaptive threshold, especially under the settings where labeled data are rare. Instead of using a uniform prior as in (Arazo et al., 2020), we use the EMA of model predictions $\tilde{p}_t$ from Eq. 6 as an estimate of the expectation of prediction distribution over unlabeled data. We optimize the cross-entropy of $\tilde{p}_t$ and $\overline{p} = \mathbb{E}_{\mu B}[p_m(y|\Omega(u_b))]$ over mini-batch as an estimate of $H(\mathbb{E}_u[p_m(y|u)])$. Considering that

the underlying pseudo label distribution may not be uniform, we propose to modulate the fairness objective in a self-adaptive way, i.e., normalizing the expectation of probability by the histogram distribution of pseudo labels to counter the negative effect of imbalance as:

$$\overline{p} = \frac{1}{\mu B} \sum_{b=1}^{\mu B} \mathbb{1} \left( \max\left(q_b\right) \geq \tau_t(\arg\max\left(q_b\right)\right) Q_b,$$

$$\overline{h} = \mathrm{Hist}_{\mu B} \left( \mathbb{1} \left( \max\left(q_b\right) \geq \tau_t(\arg\max\left(q_b\right)\right) \hat{Q}_b \right). \tag{9}$$

Similar to $\tilde{p}_t$, we compute $\tilde{h}_t$ as:

$$\tilde{h}_t = \lambda \tilde{h}_{t-1} + (1-\lambda) \mathrm{Hist}_{\mu B} \left(\hat{q}_b\right). \tag{10}$$

The self-adaptive fairness (SAF) $L_f$ at the $t$-th iteration is formulated as:

$$\mathcal{L}_f = -\mathcal{H} \left( \mathrm{SumNorm} \left( \frac{\tilde{p}_t}{\tilde{h}_t} \right), \mathrm{SumNorm} \left( \frac{\overline{p}}{\overline{h}} \right) \right), \tag{11}$$

where $\mathrm{SumNorm} = (\cdot)/\sum(\cdot)$. SAF encourages the expectation of the output probability for each mini-batch to be close to a marginal class distribution of the model, after normalized by histogram distribution. It helps the model produce diverse predictions especially for barely supervised settings (Sohn et al., 2020), thus converges faster and generalizes better. This is also showed in Figure 1(b).

The overall objective for FreeMatch at $t$-th iteration is:

$$\mathcal{L} = \mathcal{L}_s + w_u \mathcal{L}_u + w_f \mathcal{L}_f, \tag{12}$$

where $w_u$ and $w_f$ represents the loss weight for $\mathcal{L}_u$ and $\mathcal{L}_f$ respectively. With $\mathcal{L}_u$ and $\mathcal{L}_f$, FreeMatch maximizes the mutual information between its outputs and inputs. We present the procedure of FreeMatch in Algorithm 1 of Appendix.

## 5 EXPERIMENTS

### 5.1 SETUP

We evaluate FreeMatch on common benchmarks: CIFAR-10/100 (Krizhevsky et al., 2009), SVHN (Netzer et al., 2011), STL-10 (Coates et al., 2011) and ImageNet (Deng et al., 2009). Following previous work (Sohn et al., 2020; Xu et al., 2021; Zhang et al., 2021; Oliver et al., 2018), we conduct experiments with varying amounts of labeled data. In addition to the commonly-chosen labeled amounts, following (Sohn et al., 2020), we further include the most challenging case of CIFAR-10: each class has only *one* labeled sample.

For fair comparison, we train and evaluate all methods using the unified codebase TorchSSL (Zhang et al., 2021) with the same backbones and hyperparameters. Concretely, we use Wide ResNet-28-2 (Zagoruyko & Komodakis, 2016) for CIFAR-10, Wide ResNet-28-8 for CIFAR-100, Wide ResNet-37-2 (Zhou et al., 2020) for STL-10, and ResNet-50 (He et al., 2016) for ImageNet. We use SGD with a momentum of $0.9$ as optimizer. The initial learning rate is $0.03$ with a cosine learning rate decay schedule as $\eta = \eta_0 \cos(\frac{7\pi k}{16K})$, where $\eta_0$ is the initial learning rate, $k(K)$ is the current (total) training step and we set $K = 2^{20}$ for all datasets. At the testing phase, we use an exponential moving average with the momentum of $0.999$ of the training model to conduct inference for all algorithms. The batch size of labeled data is $64$ except for ImageNet where we set $128$. We use the same weight decay value, pre-defined threshold $\tau$, unlabeled batch ratio $\mu$ and loss weights introduced for Pseudo-Label (Lee et al., 2013), $\Pi$ model (Rasmus et al., 2015), Mean Teacher (Tarvainen & Valpola, 2017), VAT (Miyato et al., 2018), MixMatch (Berthelot et al., 2019b), ReMixMatch (Berthelot et al., 2019a), UDA (Xie et al., 2020a), FixMatch (Sohn et al., 2020), and FlexMatch (Zhang et al., 2021).

We implement MPL based on UDA as in (Pham et al., 2021), where we set temperature as $0.8$ and $w_u$ as $10$. We do not fine-tune MPL on labeled data as in (Pham et al., 2021) since we find fine-tuning will make the model overfit the labeled data especially with very few of them. For Dash, we use the same parameters as in (Xu et al., 2021) except we warm-up on labeled data for

Table 1: Error rates on CIFAR-10/100, SVHN, and STL-10 datasets. The fully-supervised results of STL-10 are unavailable since we do not have label information for its unlabeled data. **Bold** indicates the best result and underline indicates the second-best result. The significant tests and average error rates for each dataset can be found in Appendix E.1.

| Dataset | CIFAR-10 | | | | CIFAR-100 | | | SVHN | | | STL-10 | |
|---|---|---|---|---|---|---|---|---|---|---|---|---|
| # Label | 10 | 40 | 250 | 4000 | 400 | 2500 | 10000 | 40 | 250 | 1000 | 40 | 1000 |
| Π Model (Rasmus et al., 2015) | $79.18_{\pm1.11}$ | $74.34_{\pm1.76}$ | $46.24_{\pm1.29}$ | $13.13_{\pm0.59}$ | $86.96_{\pm0.80}$ | $58.80_{\pm0.66}$ | $36.65_{\pm0.00}$ | $67.48_{\pm0.95}$ | $13.30_{\pm1.12}$ | $7.16_{\pm0.11}$ | $74.31_{\pm0.85}$ | $32.78_{\pm0.40}$ |
| Pseudo Label (Lee et al., 2013) | $80.21_{\pm0.55}$ | $74.61_{\pm0.26}$ | $46.49_{\pm2.20}$ | $15.08_{\pm0.19}$ | $87.45_{\pm0.85}$ | $57.74_{\pm0.28}$ | $36.55_{\pm0.24}$ | $64.61_{\pm5.6}$ | $15.59_{\pm0.95}$ | $9.40_{\pm0.32}$ | $74.68_{\pm0.99}$ | $32.64_{\pm0.71}$ |
| VAT (Miyato et al., 2018) | $79.81_{\pm1.17}$ | $74.66_{\pm2.12}$ | $41.03_{\pm1.79}$ | $10.51_{\pm0.12}$ | $85.20_{\pm1.40}$ | $46.84_{\pm0.79}$ | $32.14_{\pm0.19}$ | $74.75_{\pm3.38}$ | $4.33_{\pm0.12}$ | $4.11_{\pm0.20}$ | $74.74_{\pm0.38}$ | $37.95_{\pm1.12}$ |
| MeanTeacher (Tarvainen & Valpola, 2017) | $76.37_{\pm0.44}$ | $70.09_{\pm1.60}$ | $37.46_{\pm3.30}$ | $8.10_{\pm0.21}$ | $81.11_{\pm1.44}$ | $45.17_{\pm1.06}$ | $31.75_{\pm0.23}$ | $36.09_{\pm3.98}$ | $3.45_{\pm0.03}$ | $3.27_{\pm0.05}$ | $71.72_{\pm1.45}$ | $33.90_{\pm1.37}$ |
| MixMatch (Berthelot et al., 2019b) | $65.76_{\pm7.06}$ | $36.19_{\pm6.48}$ | $13.63_{\pm0.59}$ | $6.66_{\pm0.26}$ | $67.59_{\pm0.66}$ | $39.76_{\pm0.48}$ | $27.78_{\pm0.29}$ | $30.60_{\pm8.39}$ | $4.56_{\pm0.32}$ | $3.69_{\pm0.37}$ | $54.93_{\pm0.96}$ | $21.70_{\pm0.68}$ |
| ReMixMatch (Berthelot et al., 2019a) | $20.77_{\pm7.48}$ | $9.88_{\pm1.03}$ | $6.30_{\pm0.05}$ | $4.84_{\pm0.01}$ | $42.75_{\pm1.05}$ | $\mathbf{26.03}_{\pm0.35}$ | $\mathbf{20.02}_{\pm0.27}$ | $24.04_{\pm9.13}$ | $6.36_{\pm0.22}$ | $5.16_{\pm0.31}$ | $32.12_{\pm6.24}$ | $6.74_{\pm0.14}$ |
| UDA (Xie et al., 2020a) | $34.53_{\pm10.69}$ | $10.62_{\pm3.75}$ | $5.16_{\pm0.06}$ | $4.29_{\pm0.07}$ | $46.39_{\pm1.59}$ | $27.73_{\pm0.21}$ | $22.49_{\pm0.23}$ | $5.12_{\pm4.27}$ | $\mathbf{1.92}_{\pm0.05}$ | $\mathbf{1.89}_{\pm0.01}$ | $37.42_{\pm8.44}$ | $6.64_{\pm0.17}$ |
| FixMatch (Sohn et al., 2020) | $24.79_{\pm7.65}$ | $7.47_{\pm0.28}$ | $\mathbf{4.86}_{\pm0.05}$ | $4.21_{\pm0.08}$ | $46.42_{\pm0.82}$ | $28.03_{\pm0.16}$ | $22.20_{\pm0.12}$ | $3.81_{\pm1.18}$ | $2.02_{\pm0.02}$ | $1.96_{\pm0.03}$ | $35.97_{\pm4.14}$ | $6.25_{\pm0.33}$ |
| Dash (Xu et al., 2021) | $27.28_{\pm14.09}$ | $8.93_{\pm3.11}$ | $5.16_{\pm0.23}$ | $4.36_{\pm0.11}$ | $44.82_{\pm0.96}$ | $27.15_{\pm0.22}$ | $21.88_{\pm0.07}$ | $\underline{2.19}_{\pm0.18}$ | $2.04_{\pm0.02}$ | $1.97_{\pm0.01}$ | $34.52_{\pm4.30}$ | $6.39_{\pm0.56}$ |
| MPL (Pham et al., 2021) | $23.55_{\pm6.01}$ | $6.62_{\pm0.91}$ | $5.76_{\pm0.24}$ | $4.55_{\pm0.04}$ | $46.26_{\pm1.84}$ | $27.71_{\pm0.19}$ | $21.74_{\pm0.09}$ | $9.33_{\pm8.02}$ | $2.29_{\pm0.04}$ | $2.28_{\pm0.02}$ | $35.76_{\pm4.83}$ | $6.66_{\pm0.00}$ |
| FlexMatch (Zhang et al., 2021) | $\underline{13.85}_{\pm12.04}$ | $\underline{4.97}_{\pm0.06}$ | $4.98_{\pm0.09}$ | $\underline{4.19}_{\pm0.01}$ | $\underline{39.94}_{\pm1.62}$ | $26.49_{\pm0.20}$ | $21.90_{\pm0.15}$ | $8.19_{\pm3.20}$ | $6.59_{\pm2.29}$ | $6.72_{\pm0.30}$ | $\underline{29.15}_{\pm4.16}$ | $\underline{5.77}_{\pm0.18}$ |
| FreeMatch | $\mathbf{8.07}_{\pm4.24}$ | $\mathbf{4.90}_{\pm0.04}$ | $\underline{4.88}_{\pm0.18}$ | $\mathbf{4.10}_{\pm0.02}$ | $\mathbf{37.98}_{\pm0.42}$ | $\underline{26.47}_{\pm0.20}$ | $\underline{21.68}_{\pm0.03}$ | $\mathbf{1.97}_{\pm0.02}$ | $\underline{1.97}_{\pm0.01}$ | $\underline{1.96}_{\pm0.03}$ | $\mathbf{15.56}_{\pm0.55}$ | $\mathbf{5.63}_{\pm0.15}$ |
| Fully-Supervised | $4.62_{\pm0.05}$ | | | | $19.30_{\pm0.09}$ | | | $2.13_{\pm0.01}$ | | | - | |

2 epochs since too much warm-up will lead to the overfitting (i.e. 2,048 training iterations). For FreeMatch, we set $w_u = 1$ for all experiments. Besides, we set $w_f = 0.01$ for CIFAR-10 with 10 labels, CIFAR-100 with 400 labels, STL-10 with 40 labels, ImageNet with 100k labels, and all experiments for SVHN. For other settings, we use $w_f = 0.05$. For SVHN, we find that using a low threshold at early training stage impedes the model to cluster the unlabeled data, thus we adopt two training techniques for SVHN: (1) warm-up the model on only labeled data for 2 epochs as Dash; and (2) restrict the SAT within the range $[0.9, 0.95]$. The detailed hyperparameters are introduced in Appendix D. We train each algorithm 3 times using different random seeds and report the best error rates of all checkpoints (Zhang et al., 2021).

## 5.2 QUANTITATIVE RESULTS

The Top-1 classification error rates of CIFAR-10/100, SVHN, and STL-10 are reported in Table 1. The results on ImageNet with 100 labels per class are in Table 2. We also provide detailed results on precision, recall, F1 score, and confusion matrix in Appendix E.3. These quantitative results demonstrate that FreeMatch achieves the best performance on CIFAR-10, STL-10, and ImageNet datasets, and it produces very close results on SVHN to the best competitor. On CIFAR-100, FreeMatch is better than ReMixMatch when there are 400 labels. The good performances of ReMixMatch on CIFAR-100 (2500) and CIFAR-100 (10000) are probably brought by the mix up (Zhang et al., 2017) technique and the self-supervised learning part. On ImageNet with 100k labels, FreeMatch significantly outperforms the latest counterpart FlexMatch by **1.28**%[3]. We also notice that FreeMatch exhibits fast computation in ImageNet from Table 2. Note that FlexMatch is much slower than FixMatch and FreeMatch because it needs to maintain a list that records whether each sample is clean, which needs heavy indexing computation budget on large datasets.

Noteworthy is that, FreeMatch consistently outperforms other methods by a large margin on settings with *extremely limited labeled data*: **5.78**% on CIFAR-10 with 10 labels, **1.96**% on CIFAR-100 with 400 labels, and surprisingly **13.59**% on STL-10 with 40 labels. STL-10 is a more realistic and challenging dataset compared to others, which consists of a large unlabeled set of 100k images. The significant improvements demonstrate the capability and potential of FreeMatch to be deployed in real-world applications.

Table 2: Error rates and runtime on ImageNet with 100 labels per class.

| | Top-1 | Top-5 | Runtime (sec./iter.) |
|---|---|---|---|
| FixMatch | 43.66 | 21.80 | **0.4** |
| FlexMatch | 41.85 | 19.48 | 0.6 |
| FreeMatch | **40.57** | **18.77** | **0.4** |

## 5.3 QUALITATIVE ANALYSIS

We present some qualitative analysis: Why and how does FreeMatch work? What other benefits does it bring? We evaluate the class average threshold and average sampling rate on STL-10 (40) (i.e., 40 labeled samples on STL-10) of FreeMatch to demonstrate how it works aligning with our theoretical analysis. We record the threshold and compute the sampling rate for each batch during training. The sampling rate is calculated on unlabeled data as $\frac{\sum_b^{\mu B} \mathbb{1}(\max(q_b) > \tau_t(\arg\max(q_b)))}{\mu B}$. We

---

[3]Following (Zhang et al., 2021), we train ImageNet for $2^{20}$ iterations like other datasets for a fair comparison. We use 4 Tesla V100 GPUs on ImageNet.

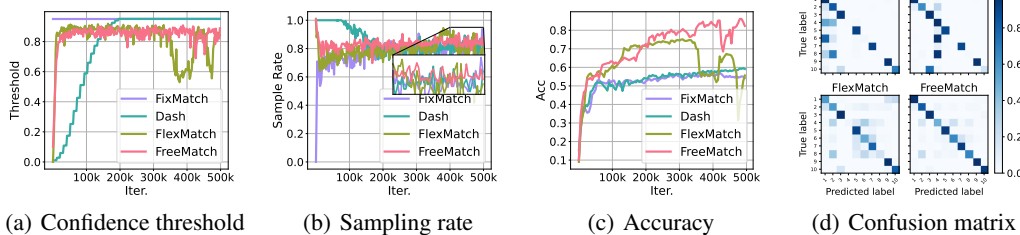

| (a) Confidence threshold | (b) Sampling rate | (c) Accuracy | (d) Confusion matrix |

Figure 3: How FreeMatch works in STL-10 with 40 labels, compared to others. (a) Class-average confidence threshold; (b) class-average sampling rate; (c) convergence speed in terms of accuracy; (d) confusion matrix, where fading colors of diagonal elements refer to the disparity of accuracy.

also plot the convergence speed in terms of accuracy and the confusion matrix to show the proposed component in FreeMatch helps improve performance. From Figure 3(a) and Figure 3(b), one can observe that the threshold and sampling rate change of FreeMatch is mostly consistent with our theoretical analysis. That is, at the early stage of training, the threshold of FreeMatch is relatively lower, compared to FlexMatch and FixMatch, resulting in higher unlabeled data utilization (sampling rate), which fastens the convergence. As the model learns better and becomes more confident, the threshold of FreeMatch increases to a high value to alleviate the confirmation bias, leading to stably high sampling rate. Correspondingly, the accuracy of FreeMatch increases vastly (as shown in Figure 3(c)) and resulting better class-wise accuracy (as shown in Figure 3(d)). Note that Dash fails to learn properly due to the employment of the high sampling rate until 100k iterations.

To further demonstrate the effectiveness of the class-specific threshold in FreeMatch, we present the t-SNE (Van der Maaten & Hinton, 2008) visualization of features of FlexMatch and FreeMatch on STL-10 (40) in Figure 5 of Appendix E.8. We exhibit the corresponding local threshold for each class. Interestingly, FlexMatch has a high threshold, i.e., pre-defined $0.95$, for class $0$ and class $6$, yet their feature variances are very large and are confused with other classes. This means the class-wise thresholds in FlexMatch cannot accurately reflect the learning status. In contrast, FreeMatch clusters most classes better. Besides, for the similar classes $1, 3, 5, 7$ that are confused with each other, FreeMatch retains a higher average threshold $0.87$ than $0.84$ of FlexMatch, enabling to mask more wrong pseudo labels. We also study the pseudo label accuracy in Appendix E.9 and shows FreeMatch can reduce noise during training.

## 5.4 ABLATION STUDY

**Self-adaptive Threshold** We conduct experiments on the components of SAT in FreeMatch and compare to the components in FlexMatch (Zhang et al., 2021), FixMatch (Sohn et al., 2020), Class-Balanced Self-Training (CBST) (Zou et al., 2018), and Relative Threshold (RT) in AdaMatch (Berthelot et al., 2022). The ablation is conducted on CIFAR-10 (40 labels).

As shown in Table 3, SAT achieves the best performance among all the threshold schemes. Self-adaptive global threshold $\tau_t$ and local threshold $\mathrm{MaxNorm}(\tilde{p}_t(c))$ themselves also achieve comparable results, compared to the fixed threshold $\tau$, demonstrating both local and global threshold proposed are good learning effect estimators. When using CPL $\mathcal{M}(\beta(c))$ to adjust $\tau_t$, the result is worse than the fixed threshold and exhibits larger variance, indicating potential instability of CPL. AdaMatch (Berthelot et al., 2022) uses the RT, which can be viewed as a global threshold at $t$-th iteration computed on the predictions of labeled data without EMA, whereas

Table 3: Comparison of different thresholding schemes.

| Threshold | CIFAR-10 (40) |
| --- | --- |
| $\tau$ (FixMatch) | $7.47_{\pm 0.28}$ |
| $\tau * \mathcal{M}(\beta(c))$ (FlexMatch) | $4.97_{\pm 0.06}$ |
| $\tau * \mathrm{MaxNorm}(\tilde{p}_t(c))$ | $5.13_{\pm 0.03}$ |
| $\tau_t$ (Global) | $6.06_{\pm 0.65}$ |
| $\tau_t * \mathcal{M}(\beta(c))$ | $8.40_{\pm 2.49}$ |
| CBST | $16.65_{\pm 2.90}$ |
| RT (AdaMatch) | $6.09_{\pm 0.65}$ |
| SAT (Global and Local) | $\mathbf{4.92}_{\pm 0.04}$ |

FreeMatch conducts computation of $\tau_t$ with EMA on unlabeled data that can better reflect the overall data distribution. For class-wise threshold, CBST (Zou et al., 2018) maintains a pre-defined sampling rate, which could be the reason for its bad performance since the sampling rate should be changed during training as we analyzed in Sec. 2. Note that we did not include $L_f$ in this ablation for a fair comparison. Ablation study in Appendix E.4 and E.5 on FixMatch and FlexMatch with different thresholds shows SAT serves to reduce hyperparameter-tuning computation or overall training time in the event of similar performance for an optimally selected threshold.

**Self-adaptive Fairness** As illustrated in Table 4, we also empirically study the effect of SAF on CIFAR-10 (10 labels). We study the original version of fairness objective as in (Arazo et al., 2020). Based on that, we study the operation of normalization probability by histograms and show that countering the effect of imbalanced underlying distribution indeed helps the model to learn and diverse better. One may notice that adding original fairness regularization alone already helps improve the performance. Whereas adding normalization operation in the log operation hurts the performance,

Table 4: Comparison of different class fairness items.

| Fairness | CIFAR-10 (10) |
|---|---|
| w/o fairness | $10.37_{\pm 7.70}$ |
| $U \log \overline{p}$ | $9.57_{\pm 6.67}$ |
| $U \log \mathrm{SumNorm}(\frac{\overline{p}}{h})$ | $12.07_{\pm 5.23}$ |
| DA (AdaMatch) | $32.94_{\pm 1.83}$ |
| DA (ReMixMatch) | $11.06_{\pm 8.21}$ |
| SAF | $\mathbf{8.07}_{\pm 4.24}$ |

suggesting the underlying batch data are indeed not uniformly distributed. We also evaluate Distribution Alignment (DA) for class fairness in ReMixMatch (Berthelot et al., 2019a) and AdaMatch (Berthelot et al., 2022), showing inferior results than SAF. A possible reason for the worse performance of DA (AdaMatch) is that it only uses labeled batch prediction as the target distribution which cannot reflect the true data distribution especially when labeled data is scarce and changing the target distribution to the ground truth uniform, i.e., DA (ReMixMatch), is better for the case with extremely limited labels. We also proved SAF can be easily plugged into FlexMatch and bring improvements in Appendix E.6. The EMA decay ablation and performances of imbalanced settings are in Appendix E.5 and Appendix E.7.

# 6 RELATED WORK

To reduce confirmation bias (Arazo et al., 2020) in pseudo labeling, confidence-based thresholding techniques are proposed to ensure the quality of pseudo labels (Xie et al., 2020a; Sohn et al., 2020; Zhang et al., 2021; Xu et al., 2021), where only the unlabeled data whose confidences are higher than the threshold are retained. UDA (Xie et al., 2020a) and FixMatch (Sohn et al., 2020) keep the fixed pre-defined threshold during training. FlexMatch (Zhang et al., 2021) adjusts the pre-defined threshold in a class-specific fashion according to the per-class learning status estimated by the number of confident unlabeled data. A co-current work Adsh (Guo & Li, 2022) explicitly optimizes the number of pseudo labels for each class in the SSL objective to obtain adaptive thresholds for imbalanced Semi-supervised Learning. However, it still needs a user-predefined threshold. Dash (Xu et al., 2021) defines a threshold according to the loss on labeled data and adjusts the threshold according to a fixed mechanism. A more recent work, AdaMatch (Berthelot et al., 2022), aims to unify SSL and domain adaptation using a pre-defined threshold multiplying the average confidence of the labeled data batch to mask noisy pseudo labels. It needs a pre-defined threshold and ignores the unlabeled data distribution especially when labeled data is too rare to reflect the unlabeled data distribution. Besides, distribution alignment (Berthelot et al., 2019a; 2022) is also utilized in Adamatch to encourage fair predictions on unlabeled data. Previous methods might fail to choose meaningful thresholds due to ignorance of the relationship between the model learning status and thresholds. Chen et al. (2020); Kumar et al. (2020) try to understand self-training / thresholding from the theoretical perspective. We use a motivating example to derive some implications and further adjust meaningful thresholds according to the learning status satisfying the derived implications.

Except consistency regularization, entropy-based regularization is also used in SSL. Entropy minimization (Grandvalet et al., 2005) encourages the model to make confident predictions for all samples disregarding the actual class predicted. Maximization of expectation of entropy (Andreas Krause, 2010; Arazo et al., 2020) over all samples is also proposed to induce fairness to the model, enforcing the model to predict each class at the same frequency. But previous ones assume a uniform prior for underlying data distribution and also ignore the batch data distribution. Distribution alignment (Berthelot et al., 2019a) adjusts the pseudo labels according to labeled data distribution and the EMA of model predictions.

# 7 CONCLUSION

We proposed FreeMatch that utilizes self-adaptive thresholding and class-fairness regularization for SSL. FreeMatch outperforms strong competitors across a variety of SSL benchmarks, especially in the barely-supervised setting. We believe that confidence thresholding has more potential in SSL. A potential limitation is that the adaptiveness still originates from the heuristics of the model prediction, and we hope the efficacy of FreeMatch inspires more research for optimal thresholding.

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

## A    EXPERIMENTAL DETAILS OF THE "TWO-MOON" DATASET.

We generate only two labeled data points (one label per each class, denoted by black dot and round circle) and 1,000 unlabeled data points (in gray) in 2-D space. We train a 3-layer MLP with 64 neurons in each layer and ReLU activation for 2,000 iterations. The red samples indicate the different samples whose confidence values are above the threshold of FreeMatch but below that of FixMatch. The sampling rate is computed on unlabeled data as $\sum_b^{N_U} \mathbb{1}(\max(q_b) > \tau)/N_U$. Results are averaged 5 times.

## B    PROOF OF THEOREM 2.1

**Theorem 2.1** *For a binary classification problem as mentioned above, the pseudo label $Y_p$ has the following probability distribution:*

$$P(Y_p = 1) = \frac{1}{2}\Phi(\frac{\frac{\mu_2-\mu_1}{2} - \frac{1}{\beta}\log(\frac{\tau}{1-\tau})}{\sigma_2}) + \frac{1}{2}\Phi(\frac{\frac{\mu_1-\mu_2}{2} - \frac{1}{\beta}\log(\frac{\tau}{1-\tau})}{\sigma_1}),$$

$$P(Y_p = -1) = \frac{1}{2}\Phi(\frac{\frac{\mu_2-\mu_1}{2} - \frac{1}{\beta}\log(\frac{\tau}{1-\tau})}{\sigma_1}) + \frac{1}{2}\Phi(\frac{\frac{\mu_1-\mu_2}{2} - \frac{1}{\beta}\log(\frac{\tau}{1-\tau})}{\sigma_2}), \quad (13)$$

$$P(Y_p = 0) = 1 - P(Y_p = 1) - P(Y_p = -1),$$

*where $\Phi$ is the cumulative distribution function of a standard normal distribution. Moreover, $P(Y_p = 0) = 0$ increases as $\mu_2 - \mu_1$ gets smaller.*

*Proof.* A sample $x$ will be assigned pseudo label 1 if

$$\frac{1}{1 + \exp\left(-\beta(x - \frac{\mu_1+\mu_2}{2})\right)} > \tau,$$

which is equivalent to

$$x > \frac{\mu_1 + \mu_2}{2} + \frac{1}{\beta}\log(\frac{\tau}{1 - \tau}).$$

Likewise, $x$ will be assigned pseudo label -1 if

$$\frac{1}{1 + \exp\left(-\beta(x - \frac{\mu_1+\mu_2}{2})\right)} < 1 - \tau,$$

which is equivalent to

$$x < \frac{\mu_1 + \mu_2}{2} - \frac{1}{\beta}\log(\frac{\tau}{1 - \tau}).$$

If we integrate over $x$, we arrive at the following conditional probabilities:

$$P(Y_p = 1|Y = 1) = \Phi(\frac{\frac{\mu_2-\mu_1}{2} - \frac{1}{\beta}\log(\frac{\tau}{1-\tau})}{\sigma_2}),$$

$$P(Y_p = 1|Y = -1) = \Phi(\frac{\frac{\mu_1-\mu_2}{2} - \frac{1}{\beta}\log(\frac{\tau}{1-\tau})}{\sigma_1}),$$

$$P(Y_p = -1|Y = -1) = \Phi(\frac{\frac{\mu_2-\mu_1}{2} - \frac{1}{\beta}\log(\frac{\tau}{1-\tau})}{\sigma_1}),$$

$$P(Y_p = -1|Y = 1) = \Phi(\frac{\frac{\mu_1-\mu_2}{2} - \frac{1}{\beta}\log(\frac{\tau}{1-\tau})}{\sigma_2}).$$

Recall that $P(Y = 1) = P(Y = -1) = 0.5$, therefore

$$P(Y_p = 1) = \frac{1}{2}\Phi(\frac{\frac{\mu_2-\mu_1}{2} - \frac{1}{\beta}\log(\frac{\tau}{1-\tau})}{\sigma_2}) + \frac{1}{2}\Phi(\frac{\frac{\mu_1-\mu_2}{2} - \frac{1}{\beta}\log(\frac{\tau}{1-\tau})}{\sigma_1}),$$

$$P(Y_p = -1) = \frac{1}{2}\Phi(\frac{\frac{\mu_2-\mu_1}{2} - \frac{1}{\beta}\log(\frac{\tau}{1-\tau})}{\sigma_1}) + \frac{1}{2}\Phi(\frac{\frac{\mu_1-\mu_2}{2} - \frac{1}{\beta}\log(\frac{\tau}{1-\tau})}{\sigma_2}).$$

Now, let's use $z$ to denote $\mu_2 - \mu_1$, to show that $P(Y_p = 0)$ increases as $\mu_2 - \mu_1$ gets smaller, we only need to show $P(Y_p = -1) + P(Y_p = 1)$ gets bigger. We write $P(Y_p = -1) + P(Y_p = 1)$ as

$$P(Y_p = 1) + P(Y_p = 1) = \frac{1}{2}\Phi(a_1 z - b_1) + \frac{1}{2}\Phi(-a_1 z - b_1) + \frac{1}{2}\Phi(a_2 z - b_2) + \frac{1}{2}\Phi(-a_2 z - b_2),$$

where $a_1 = \frac{1}{2\sigma_1}, a_2 = \frac{1}{2\sigma_2}, b_1 = \frac{\log(\frac{\tau}{1-\tau})}{\beta\sigma_1}, b_2 = \frac{\log(\frac{\tau}{1-\tau})}{\beta\sigma_2}$ are positive constants. We futher only need to show that $f(z) = \frac{1}{2}\Phi(a_1 z - b_1) + \frac{1}{2}\Phi(-a_1 z - b_1)$ is monotone increasing on $(0, \infty)$. Take the derivative of $z$, we have

$$f'(z) = \frac{1}{2}a_1(\phi(a_1 z - b_1) - \phi(-a_1 z - b_1)),$$

where $\phi$ is the probability density function of a standard normal distribution. Since $|a_1 z - b_1| < | - a_1 z - b_1|$, we have $f'(z) > 0$, and the proof is complete.

$\square$

## C  ALGORITHM

We present the pseudo algorithm of FreeMatch. Compared to FixMatch, each training step involves updating the global threshold and local threshold from the unlabeled data batch, and computing corresponding histograms. FreeMatchs introduce a very trivial computation budget compared to FixMatch, demonstrated also in our main paper.

---

**Algorithm 1** FreeMatch algorithm at $t$-th iteration.

---

1: **Input:** Number of classes $C$, labeled batch $\mathcal{X} = \{(x_b, y_b) : b \in (1, 2, \ldots, B)\}$, unlabeled batch $\mathcal{U} = \{u_b : b \in (1, 2, \ldots, \mu B)\}$, unsupervised loss weight $w_u$, fairness loss weight $w_f$, and EMA decay $\lambda$.

2: Compute $\mathcal{L}_s$ for labeled data
$\mathcal{L}_s = \frac{1}{B}\sum_{b=1}^{B} \mathcal{H}(y_b, p_m(y|\omega(x_b)))$

3: Update the global threshold
$\tau_t = \lambda\tau_{t-1} + (1-\lambda)\frac{1}{\mu B}\sum_{b=1}^{\mu B} max(q_b)$ {$q_b$ is an abbreviation of $p_m(y|\omega(u_b))$, shape of $\tau_t$: [1] }

4: Update the local threshold
$\tilde{p}_t = \lambda\tilde{p}_{t-1} + (1-\lambda)\frac{1}{\mu B}\sum_{b=1}^{\mu B} q_b$ {Shape of $\tilde{p}_t$: [C]}

5: Update histogram for $\tilde{p}_t$
$\tilde{h}_t = \lambda\tilde{h}_{t-1} + (1-\lambda)\text{Hist}_{\mu B}(\hat{q}_b)$ {Shape of $\tilde{h}_t$: [C]}

6: **for** $c = 1$ to $C$ **do**

7: $\quad\tau_t(c) = \text{MaxNorm}(\tilde{p}_t(c))\cdot\tau_t$ {Calculate SAT}

8: **end for**

9: Compute $\mathcal{L}_u$ on unlabeled data
$\mathcal{L}_u = \frac{1}{\mu B}\sum_{b=1}^{\mu B} \mathbb{1}(\max(q_b) \geq \tau_t(\arg\max(q_b)))\cdot\mathcal{H}(\hat{q}_b, Q_b)$

10: Compute expectation of probability on unlabeled data
$\overline{p} = \frac{1}{\mu B}\sum_{b=1}^{\mu B} \mathbb{1}(\max(q_b) \geq \tau_t(\arg\max(q_b)) Q_b$ {$Q_b$ is an abbr. of $p_m(y|\Omega(u_b))$, shape of $\overline{p}$: [C]}

11: Compute histogram for $\overline{p}$
$\overline{h} = \text{Hist}_{\mu B}\left(\mathbb{1}(\max(q_b) \geq \tau_t(\arg\max(q_b))\hat{Q}_b\right)$ {Shape of $\overline{h}$: [C]}

12: Compute $\mathcal{L}_f$ on unlabeled data
$\mathcal{L}_f = -\mathcal{H}\left(\text{SumNorm}(\frac{\tilde{p}_t}{\tilde{h}_t}), \text{SumNorm}(\frac{\overline{p}}{\overline{h}})\right)$

13: **Return:** $\mathcal{L}_s + w_u\cdot\mathcal{L}_u + w_f\cdot\mathcal{L}_f$

---

## D  HYPERPARAMETER SETTING

For reproduction, we show the detailed hyperparameter setting for FreeMatch in Table 5 and 6, for algorithm-dependent and algorithm-independent hyperparameters, respectively.

Table 5: Algorithm dependent hyperparameters.

| Algorithm | FreeMatch |
|---|---|
| Unlabeled Data to Labeled Data Ratio (CIFAR-10/100, STL-10, SVHN) | 7 |
| Unlabeled Data to Labeled Data Ratio (ImageNet) | 1 |
| Loss weight $w_u$ for all experiments | 1 |
| Loss weight $w_f$ for CIFAR-10 (10), CIFAR-100 (400), STL-10 (40), ImageNet (100k), SVHN | 0.01 |
| Loss weight $w_f$ for others | 0.05 |
| Thresholding EMA decay for all experiments | 0.999 |

Table 6: Algorithm independent hyperparameters.

| Dataset | CIFAR-10 | CIFAR-100 | STL-10 | SVHN | ImageNet |
|---|---|---|---|---|---|
| Model | WRN-28-2 | WRN-28-8 | WRN-37-2 | WRN-28-2 | ResNet-50 |
| Weight decay | 5e-4 | 1e-3 | 5e-4 | 5e-4 | 3e-4 |
| Batch size | 64 | | | | 128 |
| Learning rate | 0.03 | | | | |
| SGD momentum | 0.9 | | | | |
| EMA decay | 0.999 | | | | |

Note that for ImageNet experiments, we used the same learning rate, optimizer scheme, and training iterations as other experiments, and a batch size of 128 is adopted, whereas, in FixMatch, a large batch size of 1024 and a different optimizer is used. From our experiments, we found that training ImageNet with only $2^{20}$ is not enough, and the model starts converging at the end of training. Longer training iterations on ImageNet will be explored in the future. Single NVIDIA V100 is used for training on CIFAR-10, CIFAR-100, SVHN and STL-10. It costs about 2 days to train on CIFAR-10 and SVHN. 10 days are needed for the training on CIFAR-100 and STL-10.

# E  EXTENSIVE EXPERIMENT DETAILS AND RESULTS

We present extensive experiment details and results as complementary to the experiments in the main paper.

## E.1  SIGNIFICANT TESTS

We did significance test using the Friedman test. We choose the top 7 algorithms on 4 datasets (i.e., $N = 4, k = 7$). Then, we compute the F value as $\tau_F = 3.56$, which is clearly larger than the thresholds $2.661(\alpha = 0.05)$ and $2.130(\alpha = 0.1)$. This test indicates that there are significant differences between all algorithms.

To further show our significance, we report the average error rates for each dataset in Table 7. We can see FreeMatch outperforms most SSL algorithms significantly.

## E.2  CIFAR-10 (10) LABELED DATA

Following (Sohn et al., 2020), we investigate the limitations of SSL algorithms by giving only **one labeled training sample per class**. The selected 3 labeled training sets are visualized in Figure 4, which are obtained by (Sohn et al., 2020) using ordering mechanism (Carlini et al., 2019).

## E.3  DETAILED RESULTS

To comprehensively evaluate the performance of all methods in a classification setting, we further report the precision, recall, f1 score, and AUC (area under curve) results of CIFAR-10 with the same 10 labels, CIFAR-100 with 400 labels, SVHN with 40 labels, and STL-10 with 40 labels. As shown in Table 8 and 9, FreeMatch also has the best performance on precision, recall, F1 score, and AUC in addition to the top1 error rates reported in the main paper.

Table 7: The average error rates for each dataset.

|  | CIFAR-10 | CIFAR-100 | SVHN | STL-10 | Total Average |
|---|---|---|---|---|---|
| Π Model | 53.22 | 60.80 | 29.31 | 53.55 | 49.19 |
| Pseudo Label | 54.10 | 60.58 | 29.87 | 53.66 | 49.59 |
| VAT | 51.50 | 54.73 | 27.73 | 56.35 | 47.17 |
| MeanTeacher | 48.01 | 52.68 | 14.27 | 52.81 | 41.54 |
| MixMatch | 30.56 | 45.04 | 12.95 | 38.32 | 31.07 |
| ReMixMatch | 10.45 | 29.60 | 11.85 | 19.43 | 17.08 |
| UDA | 13.65 | 32.20 | 2.98 | 22.03 | 17.02 |
| FixMatch | 10.33 | 32.22 | 2.60 | 21.11 | 15.67 |
| Dash | 11.43 | 31.28 | 2.07 | 20.46 | 15.56 |
| MPL | 10.12 | 31.90 | 4.63 | 21.21 | 16.04 |
| FlexMatch | 7.00 | 29.44 | 7.17 | 17.46 | 14.40 |
| FreeMatch | **5.49** | **28.71** | **1.97** | **10.60** | **11.26** |

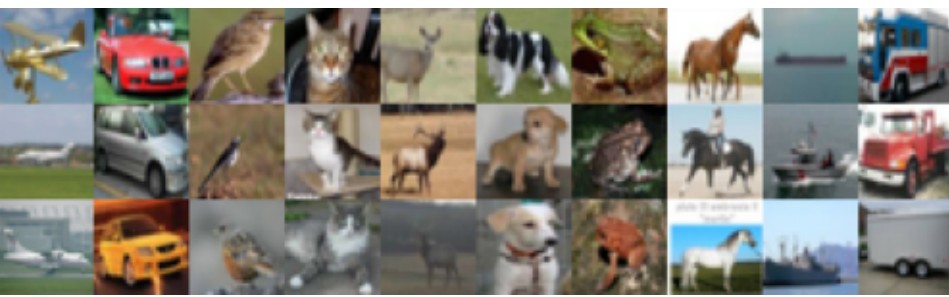

Figure 4: CIFAR-10 (10) labeled samples visualization, sorted from the most prototypical dataset (first row) to least prototypical dataset (last row).

### E.4    ABLATION OF PRE-DEFINED THRESHOLDS ON FIXMATCH AND FLEXMATCH

As shown in Table 12, the performance of FixMatch and FlexMatch is quite sensitive to the changes of the pre-defined threshold $\tau$.

### E.5    ABLATION ON EMA DECAY ON CIFAR-10 (40)

We provide the ablation study on EMA decay parameter $\lambda$ in Equation (5) and Equation (6). One can observe that different decay $\lambda$ produces the close results on CIFAR-10 with 40 labels, indicating that FreeMatch is not sensitive to this hyper-parameter. A large $\lambda$ is not encouraged since it could impede the update of global / local thresholds.

### E.6    ABLATION OF SAF ON FLEXMATCH AND FREEMATCH

In Table 13, we present the comparison of different class fairness objectives on CIFAR-10 with 10 labels. FreeMatch is better than FlexMatch in both settings. In addition, SAF is also proved effective when combined with FlexMatch.

### E.7    ABLATION OF IMBALANCED SSL

To further prove the effectiveness of FreeMatch, We evaluate FreeMatch on the imbalanced SSL setting Kim et al. (2020); Wei et al. (2021); Lee et al. (2021); Fan et al. (2021), where the labeled and the unlabeled data are both imbalanced. We conduct experiments on CIFAR-10-LT and CIFAR-100-LT with different imbalance ratios. The imbalance ratio used on CIFAR datasets is defined as $\gamma = N_{max}/N_{min}$ where $N_{max}$ is the number of samples on the head (frequent) class and $N_{min}$ the

Table 8: Precision, recall, f1 score and AUC results on CIFAR-10/100.

| Datasets | CIFAR-10 (10) | | | | CIFAR-100 (400) | | | |
|---|---|---|---|---|---|---|---|---|
| Criteria | Precision | Recall | F1 Score | AUC | Precision | Recall | F1 Score | AUC |
| UDA | 0.5304 | 0.5121 | 0.4754 | 0.8258 | 0.5813 | 0.5484 | 0.5087 | 0.9475 |
| FixMatch | 0.6436 | 0.6622 | 0.6110 | 0.8934 | 0.5574 | 0.5430 | 0.4946 | 0.9363 |
| Dash | 0.6409 | 0.5410 | 0.4955 | 0.8458 | 0.5833 | 0.5649 | 0.5215 | 0.9456 |
| MPL | 0.6286 | 0.6857 | 0.6178 | 0.7993 | 0.5799 | 0.5606 | 0.5193 | 0.9316 |
| FlexMatch | 0.6769 | 0.6861 | 0.6780 | 0.9126 | 0.6135 | 0.6193 | 0.6107 | 0.9675 |
| FreeMatch | **0.8619** | **0.8593** | **0.8523** | **0.9843** | **0.6243** | **0.6261** | **0.6137** | **0.9692** |

Table 9: Precision, recall, f1 score and AUC results on SVHN and STL-10.

| Datasets | SVHN (40) | | | | STL-10 (40) | | | |
|---|---|---|---|---|---|---|---|---|
| Criteria | Precision | Recall | F1 Score | AUC | Precision | Recall | F1 Score | AUC |
| UDA | **0.9783** | 0.9777 | 0.9780 | 0.9977 | 0.6385 | 0.5319 | 0.4765 | 0.8581 |
| FixMatch | 0.9731 | 0.9706 | 0.9716 | 0.9962 | 0.6590 | 0.5830 | 0.5405 | 0.8862 |
| Dash | 0.9782 | 0.9778 | 0.9780 | 0.9978 | 0.8117 | 0.6020 | 0.5448 | 0.8827 |
| MPL | 0.9564 | 0.9513 | 0.9512 | 0.9844 | 0.6191 | 0.5740 | 0.4999 | 0.8529 |
| FlexMatch | 0.9566 | 0.9691 | 0.9625 | 0.9975 | 0.6403 | 0.6755 | 0.6518 | 0.9249 |
| FreeMatch | **0.9783** | **0.9800** | **0.9791** | **0.9979** | **0.8489** | **0.8439** | **0.8354** | **0.9792** |

tail (rare). Note that the number of samples for class $k$ is computed as $N_k = N_{max}\gamma^{-\frac{k-1}{C-1}}$, where $C$ is the number of classes. Following (Lee et al., 2021; Fan et al., 2021), we set $N_{max} = 1500$ for CIFAR-10 and $N_{max} = 150$ for CIFAR-100, and the number of unlabeled data is twice as many for each class. We use a WRN-28-2 (Zagoruyko & Komodakis, 2016) as the backbone. We use Adam (Kingma & Ba, 2014) as the optimizer. The initial learning rate is 0.002 with a cosine learning rate decay schedule as $\eta = \eta_0 \cos(\frac{7\pi k}{16K})$, where $\eta_0$ is the initial learning rate, $k(K)$ is the current (total) training step and we set $K = 2.5 \times 10^5$ for all datasets. The batch size of labeled and unlabeled data is 64 and 128, respectively. Weight decay is set as 4e-5. Each experiment is run on three different data splits, and we report the average of the best error rates.

The results are summarized in Table 14. Compared with other standard SSL methods, FreeMach achieves the best performance across all settings. Especially on CIFAR-10 at imbalance ratio 150, FreeMatch outperforms the second best by $2.4\%$. Moreover, when plugged in the other imbalanced SSL method (Lee et al., 2021), FreeMatch still attains the best performance in most of the settings.

### E.8 T–SNE VISUALIZATION ON STL-10 (40)

We plot the T–SNE visualization of the features on STL-10 with 40 labels from FlexMatch (Zhang et al., 2021) and FreeMatch. FreeMatch shows better feature space than FlexMatch with less confusing clusters.

### E.9 PSEUDO LABEL ACCURACY ON CIFAR-10 (10)

We average the pseudo label accuracy with three random seeds and report them in Figure 6. This indicates that mapping thresholds from a high fixed threshold like FlexMatch did can prevent unlabeled samples from being involved in training. In this case, the model can overfit on labeled data and a small amount of unlabeled data. Thus the predictions on unlabeled data will incorporate

Table 10: FixMatch and FlexMatch with different thresholds on CIFAR-10 (40).

| $\tau$ | FixMatch | FlexMatch |
|---|---|---|
| 0.25 | 11.76±0.60 | 18.84±0.36 |
| 0.5 | 16.29±0.31 | 14.16±0.21 |
| 0.75 | 15.61±0.23 | 6.08±0.17 |
| 0.95 | 7.47±0.28 | 4.97±0.06 |
| 0.98 | 8.01±0.91 | 5.40±0.11 |

Table 11: Error rates of different thresholding EMA decay.

| Thresholding EMA decay | CIFAR-10 (40) |
|---|---|
| 0.9 | $4.94_{\pm 0.06}$ |
| 0.99 | $4.92_{\pm 0.08}$ |
| 0.999 | $\mathbf{4.90}_{\pm 0.04}$ |
| 0.9999 | $5.03_{\pm 0.07}$ |

Table 12: FixMatch and FlexMatch with different thresholds on CIFAR-10 (40).

| $\tau$ | **FixMatch** | **FlexMatch** |
|---|---|---|
| 0.25 | 11.76±0.60 | 18.84±0.36 |
| 0.5 | 16.29±0.31 | 14.16±0.21 |
| 0.75 | 15.61±0.23 | 6.08±0.17 |
| 0.95 | 7.47±0.28 | 4.97±0.06 |
| 0.98 | 8.01±0.91 | 5.40±0.11 |

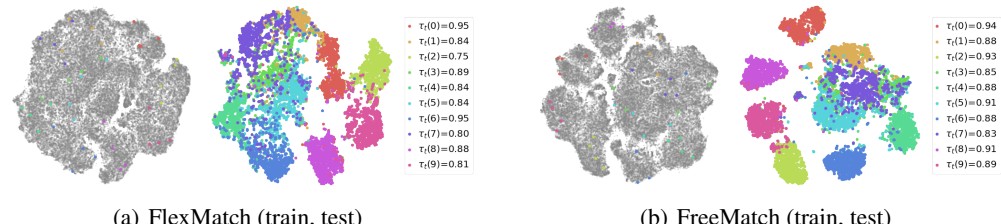

(a) FlexMatch (train, test)       (b) FreeMatch (train, test)

Figure 5: T-SNE visualization of FlexMatch and FreeMatch features on STL-10 (40). Unlabeled data is indicated by gray color. Local threshold $\tau_t(c)$ for each class is shown on the legend.

more noise. Introducing appropriate unlabeled data at training time can avoid overfitting on labeled datasets and a small amount of unlabeled data and bring more accurate pseudo labels.

### E.10   CIFAR-10 (10) CONFUSION MATRIX

We plot the confusion matrix of FreeMatch and other SSL methods on CIFAR-10 (10) in Figure 7. It is worth noting that even with the least prototypical labeled data in our setting, FreeMatch still gets good results while other SSL methods fail to separate the unlabeled data into different clusters, showing inconsistency with the low-density assumption in SSL.

Table 13: Ablation of SAF on FlexMatch and FreeMatch on CIFAR-10 (10)

| Fairness Objective | FlexMatch | FreeMatch |
|---|---|---|
| w/o SAF | 13.85±12.04 | 10.37±7.70 |
| w/ SAF | 12.60±8.16 | 8.07±4.24 |

Table 14: Error rates (%) of imbalanced SSL using 3 different random seeds.

| Dataset | CIFAR-10-LT | | CIFAR-100-LT | |
|---|---|---|---|---|
| Imbalance $\gamma$ | 50 | 150 | 20 | 100 |
| FixMatch | $18.5_{\pm 0.48}$ | $31.2_{\pm 1.08}$ | $49.1_{\pm 0.62}$ | $\mathbf{62.5}_{\pm 0.36}$ |
| FlexMatch | $17.8_{\pm 0.24}$ | $29.5_{\pm 0.47}$ | $48.9_{\pm 0.71}$ | $62.7_{\pm 0.08}$ |
| FreeMatch | $\mathbf{17.7}_{\pm 0.33}$ | $\mathbf{28.8}_{\pm 0.64}$ | $\mathbf{48.4}_{\pm 0.91}$ | $62.5_{\pm 0.23}$ |
| FixMatch w/ ABC | $14.0_{\pm 0.22}$ | $\mathbf{22.3}_{\pm 1.08}$ | $46.6_{\pm 0.69}$ | $\mathbf{58.3}_{\pm 0.41}$ |
| FlexMatch w/ ABC | $14.2_{\pm 0.34}$ | $23.1_{\pm 0.70}$ | $46.2_{\pm 0.47}$ | $58.9_{\pm 0.51}$ |
| FreeMatch w/ ABC | $\mathbf{13.9}_{\pm 0.03}$ | $\mathbf{22.3}_{\pm 0.26}$ | $\mathbf{45.6}_{\pm 0.76}$ | $58.9_{\pm 0.55}$ |

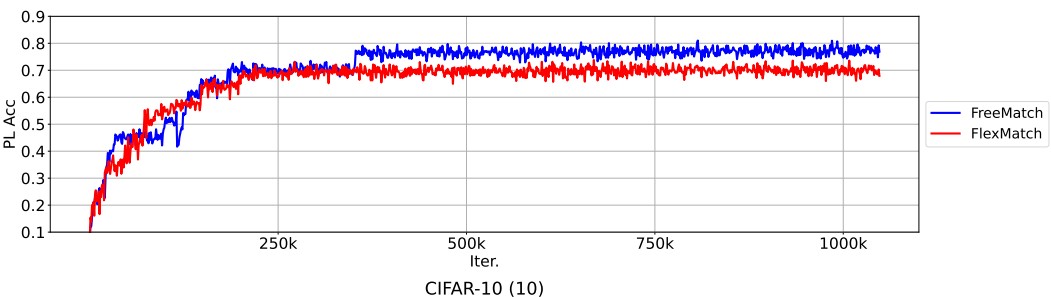

Figure 6: CIFAR-10 (10) Pseudo Label accuracy visualization.

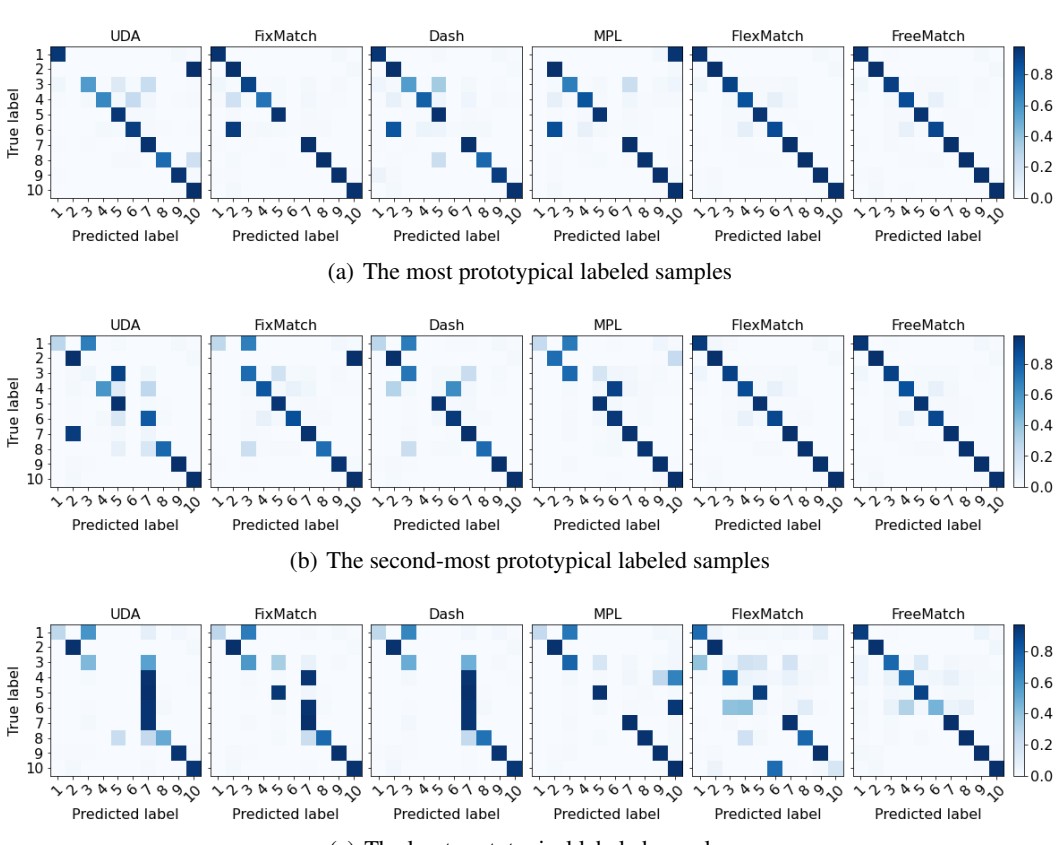

(a) The most prototypical labeled samples

(b) The second-most prototypical labeled samples

(c) The least prototypical labeled samples

Figure 7: Confusion matrix on the test set of CIFAR-10 (10). Rows correspond to the rows in Figure 4. Columns correspond to different SSL methods.

