# OpenReview forum: "FreeMatch: Self-adaptive Thresholding for Semi-supervised Learning"
_ICLR.cc/2023/Conference — ICLR 2023 poster_

### Official Review · Reviewer_axb6 · 2022-10-24

**Confidence:** 3
**Correctness:** 4
**Technical Novelty And Significance:** 3
**Empirical Novelty And Significance:** 3
**Recommendation:** 8

**Clarity, Quality, Novelty And Reproducibility:**

The adaptive thresholding scheme presented is novel to my knowledge, as is the proposed formulation for incorporation of fairness in the loss function.

**Strength And Weaknesses:**

The authors to a good job of laying out th problem and presenting their proposed approach. The approach is clean and well motivated and the empirical results are strong.

One experimental result that would have been helpful to demonstrate the importance/impact of the components of the proposed approach would have been to show performance of the loss function (12) for a fixed threshold selected by optimizing validation performance. This would help show whether the adaptive thresholding scheme is serves to reduce hyperparameter-tuning computation or overall training time (in the event of similar performance for an optimally selected threshold) or whether the adaptive threshold outperforms any fixed threshold. Additionally, this would help demonstrate the improvement from the proposed self-adaptive fairness loss term independent of the adaptive threshold.


**Summary Of The Paper:**

This paper present an adaptive approach to threshold selection for labeling of unannotated data during semi-supervised model training. The proposed approach combining adaptive threshold selection and a combination of losses leads to state of the art performance for a variety of SSL classification tasks.

**Summary Of The Review:**

Overall I'm in favor of accepting this paper as it combines novelty related to both the thresholding of unlabeled data and formulation of the loss function with empirically strong results.

---

> ### Author Response · Authors · 2022-11-11
> **Response to Reviewer axb6**
>
> Thanks for your acknowledgement in our *novelty, clear motivation, clarity and effectiveness of the proposed method, good presentation*. We see that your main concern is on more ablation experiments. We now answer your questions.
>
> **Q1**: Ablation of different fixed threshold selected by optimizing validation performance.
>
> **A1**: The following tables show the error rates of different pre-defined thresholds and those of different EMA decay used for SAT, respectively. *Note that 0.9 is quite small and 0.9999 is quite big for EMA as it changes exponentially*. We can see that the performance of FixMatch and FlexMatch is quite sensitive to the changes of the pre-defined threshold $\tau$. In fact, FlexMatch chooses $\tau=0.7$ for Imagenet rather than 0.95 to achieve better results. The tuning on $\tau$ can be very time-consuming and environmentally unfriendly if the datasets are large and complex. However, FreeMatch is not sensitive to EMA decay. **This indicates that our SAT scheme serves to reduce hyperparameter-tuning computation or overall training time (in the event of similar performance for an optimally selected threshold) and SAT outperforms any fixed threshold**. We have added the thresholding ablation in Sec 5.4 and Appendix D.4 of the revised paper.
>
> | $\tau$ | FixMatch CIFAR-10 (40) | FlexMatch CIFAR-10 (40) | | EMA decay | FreeMatch CIFAR-10 (40) |
> |--------|------------------------|-------------------------|-|-----------|-------------------------|
> | 0.25   | 11.76±0.60             | 18.84±0.36              | |    0.9    | 4.94±0.06               |
> | 0.5    | 16.29±0.31             | 14.16±0.21              | |    0.99   | 4.92±0.08               |
> | 0.75   | 15.61±0.23             | 6.08±0.17               | |    0.999  | 4.90±0.04               |
> | 0.95   | 7.47±0.28              | 4.97±0.06               | |    0.9999 | 5.03±0.07               |
> | 0.98   | 8.01±0.91              | 5.40±0.11               | |           |                         |

---

> ### Author Response · Authors · 2022-11-16
> **Looking forward to you response**
>
> Dear Reviewer axb6,
>
> Please let us know if you still have concern regarding the paper. We are looking forward to your response.

---

### Official Review · Reviewer_Ebth · 2022-10-25

**Confidence:** 5
**Clarity, Quality, Novelty And Reproducibility:** Please refer to my detailed comments …
**Correctness:** 2
**Technical Novelty And Significance:** 2
**Empirical Novelty And Significance:** 2
**Recommendation:** 5

**Strength And Weaknesses:**

Strengths:

(1) The motivation is clear.

(2) The presentation is good, and the paper is easy to follow.

(3) The proposed method is simple and effective.

Weaknesses:

(1) The main contribution lies in learning the threshold in an adaptive way. They assign various thresholds for different categories and iterations. Given the fact FlexMatch already set different weights for different classes, I think the overall novelty is unsatisfying. The contribution of self-adaptive fairness over existing strategy is also marginal.

(2) The improvements over FlexMatch on CIFAR10 and CIFAR100 are marginal.

(3) Please compare with FlexMatch under the same fairness regularization. For example, if you replace the SAF with that in FlexMatch, will the results are still better than FlexMatch?

(4) The authors claim that the proposed method can speed up the convergence. However, according to Figure 3(c), in 500k iteration, the proposed method is still not converged. Please give more explanation.

(5) For different datasets, different training strategy is adopted to improve the performance. For example, the authors define specific training requirement on STL-10, which is not consistent to the so-called self-adaptive property throughout the paper.

(6) In FixMatch and FlexMatch, the predefined threshold is very high to promise that highly-confident samples are selected for training. However, the proposed FreeMatch assigns low threshold at the beginning, which might incorporate some incorrect samples for training. Please explain this.

(7) There are several typos, such as "Eq equation 6" in page 5.


**Summary Of The Paper:**

For the semi-supervised learning task, this paper argues that existing methods might fail to utilize the unlabeled data more effectively since they either use a fixed threshold or an ad-hoc threshold adjusting scheme, resulting in inferior performance and slow convergence. Then they propose FreeMatch to adaptively adjust the confidence threshold. Experiments on several related datasets demonstrate the superiority.

**Summary Of The Review:**

The overall novelty of self-adaptive threshold learning strategy is unsatisfying. Improvement over FlexMatch on CIFAR is marginal. More experiments are suggested to support some statements.

---

> ### Author Response · Authors · 2022-11-11
> **Response to Reviewer Ebth--Part 1**
>
> Thanks for your acknowledgment of our *clear motivation, simplicity and effectiveness of the proposed method, good presentation*. We see that your main concerns are novelty, performance, and more comparisons. We now answer your questions.
>
> **Q1**: Given the fact FlexMatch already set different weights for different classes, the novelty of self-adaptive thresholding(SAT) is unsatisfying. The contribution of self-adaptive fairness(SAF) over existing strategy is also marginal.
>
> **A1**:
>
> - First, as acknowledged by Reviewer zMTV and axb6, for the thresholding part, our novelty and contribution lie in proposing SAT, which is a threshold-adjusting scheme that is free of setting thresholds *manually*. All previous thresholding-based SSL methods rely on a well-pre-defined threshold. Note that FlexMatch sets a fixed threshold manually and maps it to different classes although it claimed class-wise thresholds. **SAT is more than class-specific thresholds. It includes a self-adaptive global (dataset-specific) threshold and self-adaptive local (class-specific) thresholds. It means using SAT, we do not need to pre-define any threshold.** In addition, our local thresholds method is simpler and faster than that of FlexMatch. The following tables show the error rates of different pre-defined thresholds and those of different EMA decay used for SAT, respectively. *Note that 0.9 is quite small and 0.9999 is quite big for EMA as it changes exponentially*. We can see that the performance of FixMatch and FlexMatch is quite sensitive to the changes of the pre-defined threshold $\tau$. In fact, FlexMatch chooses $\tau=0.7$ for Imagenet rather than 0.95 to achieve better results. The tuning on $\tau$ can be very time-consuming and environmentally unfriendly if the datasets are large and complex. However, FreeMatch is not sensitive to EMA decay. **This indicates that our SAT scheme serves to reduce hyperparameter-tuning computation or overall training time (in the event of similar performance for an optimally selected threshold) and SAT outperforms any fixed threshold**. We have added the thresholding ablation in Sec 5.4 and Appendix D.4 of the revised paper.
>
> | $\tau$ | FixMatch CIFAR-10 (40) | FlexMatch CIFAR-10 (40) | | EMA decay | FreeMatch CIFAR-10 (40) |
> |--------|------------------------|-------------------------|-|-----------|-------------------------|
> | 0.25   | 11.76±0.60   | 18.84±0.36   | |    0.9    | 4.94±0.06  |
> | 0.5    | 16.29±0.31   | 14.16±0.21   | |    0.99   | 4.92±0.08 |
> | 0.75   | 15.61±0.23   | 6.08±0.17    | |    0.999  | 4.90±0.04 |
> | 0.95   | 7.47±0.28    | 4.97±0.06    | |    0.9999 | 5.03±0.07 |
> | 0.98   | 8.01±0.91    | 5.40±0.11    | |           |           |
>
> - Second, as acknowledged by *Reviewer zMTV*, we help readers understand our motivation and method more easily by discussing why thresholds should reflect the model’s learning status and providing some intuitions for designing a threshold-adjusting scheme using a motivating example.
> - Last, as acknowledged by Reviewer zMTV and axb6, the proposed novel self-adaptive fairness objective SAF can deal with the *imbalanced* influence to force the model to learn equally in all classes during training. Extensive experiments show the effectiveness of our SAT and SAF.
>
> In a nutshell, the novelty of FreeMatch lies in the easy-to-understand motivating example and the combination of novel SAT and SAF.
>
> **Q2**: The improvements over FlexMatch on CIFAR10 and CIFAR100 are marginal.
>
> **A2**: The reason that some improvements in the paper look marginal is that there exists *little room* for improvment in these settings such as CIFAR 10 with (40/250/4000), CIFAR 100 with 10000 labels. The results of FreeMatch are quite close to those of fully supervised training with 50000 labels. Besides, we report the average error rates for each dataset below. We can see FreeMatch outperforms other SSL algorithms significantly. The mean error rates for each dataset are included in Appendix D.1 in the revised paper.
>
>
> |              |     CIFAR 10 |     CIFAR 100 |     SVHN |     STL 10 |     Total Average    |
> |--------------|--------------|---------------|----------|------------|----------------------|
> | $\Pi$ Model  | 53.22  | 60.80 | 29.31 | 53.55 | 49.19 |
> | Pseudo Label | 54.10  | 60.58 | 29.87 | 53.66 | 49.59 |
> | VAT          | 51.50  | 54.73 | 27.73 | 56.35 | 47.17 |
> | MeanTeacher  | 48.01  | 52.68 | 14.27 | 52.81 | 41.54 |
> | MixMatch     | 30.56  | 45.04 | 12.95 | 38.32 | 31.07 |
> | ReMixMatch   | 10.45  | 29.60 | 11.85 | 19.43 | 17.08 |
> | UDA          | 13.65  | 32.20 | 2.98  | 22.03 | 17.02 |
> | FixMatch     | 10.33  | 32.22 | 2.60  | 21.11 | 15.67 |
> | Dash         | 11.43  | 31.28 | 2.07  | 20.46 | 15.56 |
> | MPL          | 10.12  | 31.90 | 4.63  | 21.21 | 16.04 |
> | FlexMatch    | 7.00   | 29.44 | 7.17  | 17.46 | 14.40 |
> | FreeMatch    | **5.49** | **28.71** | **1.97** | **10.60**  | **11.26**|

---

> > ### Author Response · Authors · 2022-11-11
> > **Response to Reviewer Ebth--Part 2**
> >
> > **Q3**: Compare with FlexMatch under the same fairness regularization (e.g., SAF).
> >
> > **A3**: We present the comparison of different class fairness objectives on CIFAR-10 with 10 labels. FreeMatch is better than FlexMatch in both settings. In addition, SAF is also proved effective when combined with FlexMatch. We added this ablation in Appendix D.6 of the revised paper.
> >
> > | Fairness Objective| FlexMatch | FreeMatch |
> > |-------------------|----------------|---------------|
> > | w/o SAF  | 13.85±12.04  | 10.37±7.70 |
> > | w/ SAF   | 12.60±8.16  |   8.07±4.24   |
> >
> > **Q4**: According to Figure 3c in 500k iteration, the proposed method is still not converged. Please give more explanation.
> >
> > **A4**: The accuracy of FreeMatch nearly does not increase after 400k iterations. The convergence curve looks steep because the scale is too small. As Figure 3c shows, FreeMatch achieves higher accuracy with fewer iterations and is steady than FlexMatch.
> >
> > **Q5**: For different datasets, different training strategy is adopted to improve the performance. (e.g., STL-10)
> >
> > **A5**: For different datasets, the optimal hyper-parameters are usually different. We present all the used hyper-parameters in Appendix C. Note that previous thresholding methods usually adopt different training strategies for different datasets. For example, Dash warms up the model on only labeled data and restricts the threshold. UDA, FixMatch, and FlexMatch tune thresholds for more complex datasets(e.g., ImageNet).
> >
> > **Q6**: In FixMatch and FlexMatch, the predefined threshold is very high to promise that highly-confident samples are selected for training. However, the proposed FreeMatch assigns low threshold at the beginning, which might incorporate some incorrect samples for training. Please explain this.
> >
> > **A6**: Incorporating some incorrect samples can be inevitable in the scenario of SSL. However, our SAT defines the thresholds both globally and locally according to the model learning status while FlexMatch only maps a high fixed threshold to different classes. **In some cases, mapping thresholds from a high fixed threshold can prevent unlabeled samples from being involved in training. In this case, the model can overfit on labeled data and a small amount of unlabeled data. Thus the predictions on unlabeled data can incorporate more noise. Introducing appropriate unlabeled data at training time can avoid overfitting on labeled datasets and a small amount of unlabeled data and bring more accurate pseudo labels.** For example, FlexMatch chooses threshold 0.7 for ImageNet for better performance. We average the pseudo label accuracy with three random seeds and report them below. A more detailed figure can be found in Appendix D.9 of the revised paper.
> >
> > |           | 0~250k iters | 250k~500k iters | 500k~750k iters | 750k~1000k  iters|
> > |-----------|--------|-----------|-----------|------------|
> > | FlexMatch's PL Acc on CIFAR-10 (10) | 54.26  | 69.30     | 69.70     | 70.09      |
> > | FreeMatch's PL Acc on CIFAR-10 (10) | 54.83  | 74.19     | 76.72     | 77.24      |
> >
> > **Q7**: There are several typos, such as "Eq equation" on page 5.
> >
> > **A7**: We have fixed the typos in the revised paper.

---

> ### Author Response · Authors · 2022-11-16
> **Looking forward to discuss more**
>
> Dear Reviewer Ebth,
>
> Thanks for your valuable suggestions about our paper. If our response addressed your concern, please consider raising score. Please let us know if you have further questions and we would be looking forward to discuss them in more detail.

---

### Official Review · Reviewer_zMTV · 2022-10-27

**Confidence:** 4
**Correctness:** 3
**Technical Novelty And Significance:** 3
**Empirical Novelty And Significance:** 3
**Recommendation:** 8

**Clarity, Quality, Novelty And Reproducibility:**

The paper is well written with minor linguistic errors. The idea of the threshold adjustments is first presented through a simple binary conditional gaussian example which helps easily understand the concept and the context. The structure of the paper is well organized and the content is easy to follow.

Self Adaptive Threshold (SAT), Self Adaptive Fairness, and the overall FreeMatch approach are the major novelty of this work.

The experiments (plus setup) were thorough and detailed. As the code has been shared, it is expected that the reproduction of the results is feasible.

**Strength And Weaknesses:**

Strength: Usually, semi-supervised learning (SSL) methods use either some predefined or ad-hoc threshold adjustment mechanisms  which are challenged  by FreeMatch through the idea of Self Adaptive Threshold (SAT). FreeMatch provides an automated iterative unlabelled data sampling methodology using  threshold adjustments based on the learning status of the model and using  prediction confidence scores at the latest. The methodology also ensures class  fairness and learns a generalized solution (in the context of classification).

FreeMatch has been tested on CIFAR-10/100, SVHN, STL-10, and ImageNet benchmarks and compared against existing techniques such as FixMatch, ReMixMatch, and FlexMatch. Reported results are found to be encouraging.

Weakness: Although the reported results in section 5 look promising, they lack some statistical tests. It is suggested that authors perform proper statistical tests to justify whether the achieved gains are (statistically) significant or not.  If we zoom in to the results (Table 1 and 2) we find that gains are marginal for larger (such as CIFAR-100, ImageNet) and more complex (ImageNet) datasets when compared to smaller and simpler datasets such as CIFAR-10 (or data with less number of classes). This may have exhibited scalability limitations of the proposed approach.

Also, it is surprising that for smaller datasets (such as CIFAR-10) the reported results are sometimes even better than models trained in a fully supervised fashion.



**Summary Of The Paper:**

In this paper the authors have proposed FreeMatch, a novel Semi Supervised Learning (SSL) technique that samples unlabeled data (the goal is to add pseudo labels and use them as supplementary data for model learning) through a Self Adapting (confidence) Thresholding (SAT) mechanism. The hypothesis is that an adaptive threshold (depending on the status of the model) is likely to be more effective than a predefined or ad-hoc threshold,  usually used by existing techniques.

At each training iteration, FreeMatch adjusts confidence thresholds (for each class in the context of a classification problem) by leveraging model prediction information from the corresponding step. SAT first estimates a global threshold, then modulates via the local class-specific thresholds, estimated as the Exponential Moving Average (EMA) of the probability for each class. The threshold is kept low at the beginning of the training process to accept as many weak(possibly correct) samples as possible. As the model becomes more and more confident in subsequent iterations, the threshold is gradually increased to identify and reject incorrect samples (which reduces confirmation bias).

The proposed model has been tested on CIFAR-10/100, SVHN, STL-10, and ImageNet benchmarks and the results look to be encouraging.


**Summary Of The Review:**

I have gone through the paper more than once including the appendices. Overall, the idea is quite sound, well articulated through the document, which is found easy to follow. The experiment is thorough and the reported results look encouraging. Self Adaptive Threshold (SAT), Self Adaptive Fairness, and the FreeMatch approach as a whole may benefit the SSL research.

---

> ### Author Response · Authors · 2022-11-11
> **Response to Reviewer zMTV**
>
> Thanks for your acknowledgment of our *novelty, technical soundness, thorough experiments, good reproducibility, and well-structured presentation*. We see that your main concern is the lack of some statistical tests. We now answer your question.
>
> **Q1**:  Lack of some statistical tests.
>
> **A1**:
> - The statistical tests are included in Appendix D.1 in the revised paper. We did a significance test using the Friedman test. We choose the top 7 algorithms on 4 datasets (i.e., $N=4, k=7$). Then, we compute the F value as $\tau_F=3.56$, which is clearly larger than the thresholds $2.661(\alpha=0.05)$ and $2.130(\alpha=0.1)$. This test indicates that *there are significant differences between all algorithms.*
> - Besides, the reason that some improvements in the paper look marginal is that there exists **little room** for improvement in these settings such as CIFAR 10 with 40/250/4000 labels, and CIFAR 100 with 10000 labels. The results of FreeMatch are quite close to those of fully supervised training with 50000 labels. To further show our significance, we report the average error rates for each dataset below. We can see FreeMatch outperforms other SSL algorithms significantly.
>
>
> |              |     CIFAR 10 |     CIFAR 100 |     SVHN |     STL 10 |     Total Average    |
> |--------------|--------------|---------------|----------|------------|----------------------|
> | $\Pi$ Model  | 53.22        | 60.80         | 29.31    | 53.55      | 49.19                |
> | Pseudo Label | 54.10        | 60.58         | 29.87    | 53.66      | 49.59                |
> | VAT          | 51.50        | 54.73         | 27.73    | 56.35      | 47.17                |
> | MeanTeacher  | 48.01        | 52.68         | 14.27    | 52.81      | 41.54                |
> | MixMatch     | 30.56        | 45.04         | 12.95    | 38.32      | 31.07                |
> | ReMixMatch   | 10.45        | 29.60         | 11.85    | 19.43      | 17.08                |
> | UDA          | 13.65        | 32.20         | 2.98     | 22.03      | 17.02                |
> | FixMatch     | 10.33        | 32.22         | 2.60     | 21.11      | 15.67                |
> | Dash         | 11.43        | 31.28         | 2.07     | 20.46      | 15.56                |
> | MPL          | 10.12        | 31.90         | 4.63     | 21.21      | 16.04                |
> | FlexMatch    | 7.00         | 29.44         | 7.17     | 17.46      | 14.40                |
> | FreeMatch    | **5.49**     | **28.71**     | **1.97** | **10.60**  | **11.26**            |

---

> ### Author Response · Authors · 2022-11-16
> **Looking forward to you response**
>
> Dear Reviewer zMTV,
>
> If you still have concern regarding the paper, please do not hesitate to let us know and we are looking forward to discuss them in more detail with you during the discussion period.

---

### Author Response · Authors · 2022-11-27
**Organize Discussion**

Dear senior area chairs and area chairs, could you please help us organize the discussion since the we have replied the reviewers' questions for a long time and the deadline of rebuttal is approaching?

---

### Decision · Program_Chairs · 2023-01-20

**Decision:**

Accept: poster

**Justification For Why Not Higher Score:**

- Key ideas are specific to the semi-supervised learning setting and may not be of broader interest (e.g. no large improvements)

**Justification For Why Not Lower Score:**

- Proposed method is novel and well-motivated
- Empirical evaluations show it is effective on a range of datasets


**Metareview: Summary, Strengths And Weaknesses:**

This paper proposes a semi-supervised learning algorithm that uses adaptive thresholds to select pseudolabels to be used during training. Reviewers generally appreciated the well-motivated idea, promising empirical results and clear presentation. Two reviewers thought the adaptive thresholding concept was sufficiently novel while the remaining reviewer had concerns about limited novelty and marginal improvements. Reviewers also had concerns about the importance of the proposed components. The authors provided a response that the AC thinks largely addresses these concerns. Authors are also encouraged to discuss recent work also proposing adaptive thresholding in the final version of the paper: Class-Imbalanced Semi-Supervised Learning with Adaptive Thresholding, ICML 2022.

**Note From Pc:**

if the above contains the word "oral" or "spotlight" please see: "oral" presentation means -> notable-top-5% and "spotlight" means -> notable-top-25%. As stated in our emails, we are disassociating presentation type from AC recommendations